# Diversity in the Circulation of Influenza A(H3N2) Viruses in the Northern Hemisphere in the 2018–19 Season

**DOI:** 10.3390/vaccines9040375

**Published:** 2021-04-13

**Authors:** Rakefet Pando, Shahar Stern, Ital Nemet, Aharona Glatman-Freedman, Hanna Sefty, Neta S. Zuckerman, Yaron Drori, Nehemya Friedman, John W. McCauley, Lital Keinan-Boker, Ella Mendelson, Rodney S. Daniels, Michal Mandelboim

**Affiliations:** 1The Israel Center for Disease Control, Israel Ministry of Health, Tel-Hashomer, Ramat-Gan 52621, Israel; akefet.Pando@sheba.health.gov.il (R.P.); aharona.freedman@moh.gov.il (A.G.-F.); Hanna.Sefty@MOH.GOV.IL (H.S.); Neta.Zuckerman@sheba.health.gov.il (N.S.Z.); Lital.Keinan2@MOH.GOV.IL (L.K.-B.); 2Chaim Sheba Medical Center, Central Virology Laboratory, Ministry of Health, Ramat-Gan 52621, Israel; sternyosha@mail.tau.ac.il (S.S.); ital.nemet@sheba.health.gov.il (I.N.); Yaron.Drori@sheba.health.gov.il (Y.D.); Nehemya.Friedman@sheba.health.gov.il (N.F.); Ella.Mendelson@sheba.health.gov.il (E.M.); 3Department of Epidemiology and Preventive Medicine, Sackler Faculty of Medicine, School of Public Health, Tel-Aviv University, Tel-Aviv 6997801, Israel; 4Worldwide Influenza Center, The Francis Crick Institute, London NW1 1AT, UK; John.McCauley@crick.ac.uk (J.W.M.); Rod.Daniels@crick.ac.uk (R.S.D.); 5School of Public Health, University of Haifa, Haifa 3498838, Israel

**Keywords:** influenza A, circulation, clade, vaccine

## Abstract

While vaccination is considered the most effective means to prevent influenza infection, its seasonal effectiveness varies, depending on the circulating influenza strains. Here, we characterized the circulation of influenza strains in October-2018 and March-2019 around the world. For this, we used nasopharyngeal samples collected from outpatient and hospitalized patients in Israel and data reported in ECDC, CDC, and WHO databases. Influenza A(H3N2) was dominant in Israel, while in Europe, Asia, and USA, A(H1N1)pdm09 virus circulated first, and then the A(H3N2) virus also appeared. Phylogenetic analysis indicated that A(H3N2) viruses circulating in Israel belonged to clade-3C.3a, while in Europe, Asia, and USA, A(H3N2) viruses belonged to subclade-3C.2a1, but were later replaced by clade-3C.3a viruses in USA. The vaccine A(H3N2) components of that year, A/Singapore/INFIMH-16-0019/2016-(H3N2)-like-viruses, belonged to clade-3C.2a1. The circulation of different influenza subtypes and clades of A(H3N2) viruses in a single season highlights the need for universal influenza vaccines.

## 1. Introduction

Influenza viruses cause contagious respiratory disease [1] and are classified as types A, B, and C, with the first two significantly impacting human health each year [2]. Type A viruses are divided into subtypes determined by their hemagglutinins (HA) and neuraminidases (NA), with A(H3N2) and A(H1N1)pdm09 subtypes currently circulating in humans [1,3]. While in most years a single subtype predominates, the two type A virus subtypes can co-circulate [4,5]. Similarly, there are two lineages of type B viruses—B/Yamagata and B/Victoria—which also co-circulate, but with one lineage predominating [6,7], although in some seasons, circulation of type B viruses is negligible [8]. However, influenza viruses undergo genetic change in the course of their evolution, markedly so in the genes encoding HA and NA with resultant amino acid substitutions permitting escape from host immune responses, a process termed antigenic drift, which results in recurrent seasonal influenza outbreaks/epidemics [9,10,11].

Vaccination is considered the most effective way of preventing influenza infection [12,13,14]. The composition of the annual vaccines follows recommendations developed by the World Health Organization (WHO), based on worldwide surveillance conducted by the Global Influenza Surveillance and Response System (GISRS) [13]. Recommendations are announced each year at the end of February for the northern hemisphere and in September for the southern hemisphere [15]. For the 2017–18 season, the vaccine virus recommendations for the Northern hemisphere were A/Michigan/45/2015 (H1N1) pdm09-like; A/Hong Kong/4801/2014 (H3N2)-like; B/Brisbane/60/2008-like (B/Victoria/2/87 lineage); and B/Phuket/3073/2013-like (B/Yamagata/16/88 lineage) [16]. For the 2018–19 season, two vaccine component changes were recommended to an A/Singapore/INFIMH-16-0019/2016 (H3N2)-like virus and a B/Colorado/06/2017-like virus (B/Victoria/2/87 lineage) [17].

Since 2009, A(H3N2) viruses have undergone significant genetic drift (new clades and subclades), with some associated antigenic drift, that has led to multiple changes in the recommended A(H3N2) vaccine virus over the past 12 years [15]. In recent winter seasons, the majority of A(H3N2) viruses had belonged to the HA phylogenetic subclade-3C.2a1b; however, the number of clade-3C.3a viruses increased substantially from November 2018 in several geographic regions [18,18].

The aim of this study was to review what influenza strains circulated around the world in 2018–19 and to investigate whether they resembled those used in the vaccine. We review influenza circulation in Israel, Europe, Asia, and USA over the 2018–2019 winter, with focus on the unusual pattern observed in Israel. A(H3N2) viruses of clade-3C.3a predominated in Israel from the start of the season, while the same season in Europe, Asia, and USA was biphasic, with A(H1N1) pdm09 viruses dominating for most of the season, with co-circulation of subclade-3C.2a1 A(H3N2) viruses, followed by a second phase, which was dominated by clade-3C.3a A(H3N2) viruses. Consequently, most of the A(H3N2) viruses in circulation differed from the subclade-3C.2a1, A/Singapore/INFIMH-16-0019/2016-(H3N2)-like, vaccine viruses.

## 2. Materials and Methods

### 2.1. Ethics Approval

Respiratory samples from hospitalized and non-hospitalized patients with clinical symptoms of influenza-like-illness (ILI) were analyzed for the presence of respiratory viruses as part of routine testing performed in the Sheba Medical Centre (SMC), Israel. The SMC institutional review board (IRB) approved this research (Helsinki Number 1967-15-SMC). Informed consent was not required for this study.

### 2.2. Sample Collection

Nasopharyngeal samples of patients presenting with ILI were collected into ∑-Virocult (mwe, Manchester, UK) virus transport medium. Samples (*n* = 1487) were collected between October 2018 and March 2019 in outpatient clinics located in different geographic parts of Israel as part of annul surveillance conducted by the Israel Influenza Surveillance Network (IISN). In addition, samples were collected from all hospitalized patients at the SMC (*n* = 7310) that suffered from ILI in 2018–19.

### 2.3. Nucleic Acid Extraction and Detection of Influenza Viruses

Viral RNA (vRNA) was extracted from all samples using MagNA PURE 96 (Roche, Mannheim, Germany), following the manufacturer’s instructions. All samples were then tested for the presence of influenza viruses (types A and B) and respiratory syncytial virus (RSV), in a single real-time reverse transcriptase polymerase chain reaction (RT-PCR) performed using Ambion Ag-Path master mix (Thermo Fisher Scientific, Waltham, MA, USA) and TaqMan Chemistry (qRT-PCR), on an ABI 7500 platform, as described previously [19].

### 2.4. Phylogenetic Bayesian Evolutionary Analysis by Sampling Trees (BEAST) Analysis

Influenza A(H3N2) HA gene-specific RT-PCR primers were used to amplify 1701 base-pair fragments, as described in WHO protocols [20]. The amplified products were then sequenced using ABI PRISM Dye Deoxy Terminator cycle sequencing kits (Applied Biosystems, Foster City, CA, USA) and Applied Biosystems model 3100 DNA automatic sequencing systems. The Sequencher^®^ 5.0 program (Gencodes Corporation, Ann Arbor, MI, USA) was used to prepare FASTA files for further analysis.

The evolutionary relationships for the influenza A(H3N2) HA sequences were inferred using the general time-reversible model, with proportion of invariable sites and gamma plus invariant sites-distributed rate heterogeneity (GTR+G+I), chosen using JModelTest [21]. A Bayesian Markov chain Monte Carlo (MCMC) method was run for 10 million iterations, with a relaxed clock and 10% burn-in period, with samples saved every 10,000 iterations, using BEAST version 1.8 [22]. Phylogenetic trees were visualized in FigTree version 1.4.3 (http://tree.bio.ed.ac.uk/software/figtree/ (accessed on 30 March 2020)).

A random selection of 259 influenza A(H3N2) HA gene sequences from Israeli, European, Asian, and USA samples for the years 2009–2019 used in this analysis were downloaded from the EpiFlu^TM^ database of the Global Initiative on Sharing All Influenza Data (GISAID) (www.gisaid.org (accessed on 30 March 2020)). GISAID sequences used in this manuscript are listed in Appendix A.

### 2.5. Data Distribution from Europe, Asia, and USA

Data from seasonal surveillance of influenza in Europe, Asia, and USA were collected from open-access databases. Data for 2018–19 weekly surveillance results from Asia were downloaded from WHO FluNet (https://www.who.int/influenza/gisrs_laboratory/flunet/en/ (accessed on 30 March 2020)). The data were deposited by National Influenza Centers (NICs) of the GISRS and other national influenza reference laboratories collaborating actively with GISRS.Data for 2018–19 weekly surveillance in the WHO European Region were obtained from The European Surveillance System (TESSy) database which is operaed by the European Center for Disease Control (ECDC): (https://zfs.ecdc.europa.eu/adfs/ls/?wa=wsignin1.0&wtrealm=https%3a%2f%2ftessy.ecdc.europa.eu%2fTessyWeb%2f&wctx=rm%3d0%26id%3dpassive%26ru%3d%252fTessyWeb%252f&wct=2019-07-02T06%3a28%3a43Z (accessed on 30 March 2020)).

For USA data were downloaded for2018–19 weekly surveillance from CDC-FluView (https://gis.cdc.gov/grasp/fluview/fluportaldashboard.html (accessed on 30 March 2020)), which collects data from both the U.S. WHO Collaborating Laboratories and the National Respiratory and Enteric Virus Surveillance System (NREVSS).

The weekly influenza-positive results from Europe, Asia and USA were summarized and analyzed.

### 2.6. Clade Classification from Israel, Europe, Asia, and USA

All A(H3N2) HA gene sequences that encoded full-length HA glycoprotein for viruses collected between October 2018 and March 2019, deposited in the EpiFlu^TM^ database of GISAID as of 1 July 2019, were analyzed to determine the HA gene clade and subclade of each virus. In total, 100 sequences from Israel, 2275 from Europe, 1436 from Asia, and 2447 from the USA were downloaded. Clade designation was as used in the phylogenetic analyses presented at WHO vaccine composition meetings [23].

### 2.7. Statistical Analysis

The Pearson chi-squared test was applied to evaluate the differences in positive percent between the compared groups—influenza infections in hospitalized and non-hospitalized patients, as described in Figure 1; *p* < 0.05 was considered statistically statistically significant.

## 3. Results

### 3.1. Influenza Type/Subtype Detections in Outpatients and Hospitalized Patients in Israel, Winter 2018–19

Influenza infection among both outpatients (non-hospitalized) and hospitalized patients in SMC was mainly caused by A(H3N2) viruses (76% of outpatients and 66% of hospitalized patients). Influenza A(H1N1) pdm09 viruses were also detected in both patient populations, but at a significantly higher level in hospitalized patients (33% in hospitalized patients and 23% in outpatients) (*p* < 0.0001). Influenza type B infection was infrequent (<1%) (Figure 1).

### 3.2. Genetic Changes in A(H3N2) Viruses in the 2018–19 and Preceding Influenza Seasons

To examine the compatibility of circulating strains and vaccine strains, a BEAST phylogenic analysis of representative Israeli, European, Asian, and USA samples from the 2018–19 winter, together with samples from the preceding 10 years, and the corresponding vaccine strains was performed (Figure 2). The analysis showed that the circulating viruses in Israel in 2018–19 belonged to clade-3C.3a, while the A/Singapore/INFIMH-16-0019/2016-like vaccine virus belonged to clade-3C.2a, subclade-3C.2a1. The dominant Israeli 2018–19 viruses were genetically closer to the A/Switzerland/9715293/2013-like clade-3C.3a vaccine virus used in the 2015–16 influenza season. A negligible number of 2018–19 winter samples belonged to clade-3C.2a. Two samples collected at the end of the 2017–18 season contained A(H3N2) clade-3C.3a viruses. As seen with the Israeli sequences, the USA sequences from the end of the 2018–19 winter season (February–March) belonged to clade-3C.3a. The other non-Israeli sequences from the previous decade were distributed among different clades.

### 3.3. Variability of Influenza Circulation in Israel, Europe, Asia, and USA, Winter 2018–19

In Israel, analysis of influenza A-positive samples (*n* = 597) revealed co-circulation of A(H1N1) pdm09 and A(H3N2) virus subtypes each week, with a clear A(H3N2) dominance in almost every week (Figure 3A). In contrast, analysis of influenza A-positive samples in Europe (*n* = 15,956) and Asia (*n* = 99,234) found A(H1N1)pdm09 dominance for most of the season. However, in some weeks, there was no clear dominance, and towards the end of the season, A(H3N2) dominated (Figure 3B—Europe, Figure 3C—Asia).

The pattern of influenza type A detections (*n* = 37,806) in the USA differed significantly from those seen in both Israel and Europe, with a clear A(H1N1)pdm09 dominance early in the season and a gradual increase in A(H3N2) virus circulation from week 2/2019 (Figure 3D). From week 9/2019 until the end of the season, A(H3N2) viruses predominated in the USA.

### 3.4. Comparison of Influenza A(H3N2) Clade Circulation in Israel, Europe, Asia, and USA, Winter 2018–19

Figure 4 presents the clade distribution of influenza A(H3N2) viruses in Israel, Europe, Asia and USA in the 2018–19 season. In Israel, the dominant strains throughout the winter season belonged to clade-3C.3a, with very few samples belonging to subclade-3C.2a1b and none to subclade-3C.2a2 (Figure 4A). In contrast, circulation in Europe (Figure 4B) included A(H3N2) viruses from three different genetic groups, with subclade-3C.2a.1b being dominant (>90% of cases) at the beginning of the winter and declining to less than 80% by the end of the season. The decreased circulation of subclade-3C.2a1b was accompanied by an increased incidence of clade-3C.3a viruses, which were absent at the beginning of the winter and rose to >20% by its end, with subclade-3C.2a2 viruses representing a low proportion of the A(H3N2) viruses which decreased over the course of the season. A(H3N2) clade distribution in Asia was similar to the distribution in Europe, but with different dynamics (Figure 4C). Subclade-3C.2a.1b was the dominant group throughout the winter season. The incidence of clade 3C.3a gradually increased from 0% at the beginning of winter to 24% by the end, accompanied by a decrease in subclade-3C.2a.1b, while subclade-3C.2a2 viruses presented low percentages throughout the season (Figure 4C).

In the USA, all three virus clades were detected at the beginning of the season, with significant dominance of subclade-3C.2a1b. From November, the prevalence of 3c.2a1b and 3C.2a2 subclades decreased significantly and dropped to below 10% from January until the end of the season, while the proportion of clade-3C.3a viruses increased dramatically, reaching over 90% by the end of the season (Figure 4D).

## 4. Discussion

Here, we report on atypical influenza circulation during the 2018–19 season. This season was considered unusual due to the co-circulation of both seasonal influenza subtypes, A(H3N2) and A(H1N1) pdm09, with no clear dominance in many countries (e.g., France, Germany and Spain) [24]. In addition, A(H3N2) viruses in Israel, Europe, Asia, and USA showed quite different patterns of dominance.

In recent years, differences in influenza virus circulation patterns have been reported, including summer outbreaks [25,26] and co-infection with two influenza viruses in a single patient [27]. These reports align with the fact that the patterns of annual circulation of influenza viruses have been difficult to predict in recent years.

In Israel, a comparison of outpatient (non-hospitalized) versus hospitalized patient influenza infections revealed that while subtype A(H3N2) was dominant in both patient populations, a higher percentage of hospitalized patient samples carried subtype A(H1N1) pdm09. Previous reports showed that influenza A(H1N1pdm09) infection is associated with more severe clinical features and higher intensive care unit admission rate, as compared to subtype A(H3N2) infection [28,29], which might explain the higher A(H1N1) pdm09-infection rate in hospitalized patients. 

As it is located in the northern hemisphere, Israeli influenza detections during winter seasons usually align with those seen in Europe, Asia, and USA [30] However, analysis of the 2018–19 winter data demonstrated that while A(H3N2) clade-3C.3a viruses were dominant throughout Israel, viruses in this clade were detected in increasing proportions as the season progressed in Europe, Asia, and USA, becoming dominant in USA by the end of the season. However, since influenza circulation in Europe is complex, with different patterns reported for individual regions and countries, it is important to study each country’s data individually. Segaloff et al. recently reported that in several European countries (e.g., Belarus, Belgium, and Turkey) A(H3N2) viruses were, as in Israel, dominant throughout the winter season (but mainly subclade-3C.2a1b viruses were detected), while in other European countries, A(H1N1)pdm09 dominance or A(H1N1)pdm09-A(H3N2) co-dominance was reported [24].

BEAST phylogenetic analysis of Israeli samples from 2009-2019 showed that the majority of 2018–19 A(H3N2) detections belonged to clade-3C.3a, while the 2018–19 A/Singapore/INFIMH-16-0019/2016-like vaccine virus belonged to subclade-3C.2a1; few subclade-3C.2a1 viruses were detected among Israeli patients with ILI symptoms in 2018–19. Such disparity between circulating viruses and the vaccine strain could have resulted in low immunity of populations against clade-3C.3a viruses, allowing significant spread of clade-3C.3a viruses in Israel, Asia, USA, and some countries in Europe, leading to reduced vaccine effectiveness (VE). Indeed, in Israel, we found a very low VE against A(H3N2) influenza virus in the winter of 2018–19 among infants, children, and young adults though, among adults aged ≥45 years, we found higher VE which could be due to previous exposure to clade-3C.3a viruses a few years earlier [31]. Similar findings were also found in Europe and in the USA, where clade-3C.3a viruses had circulated at high levels in recent years [32,33]. In an attempt to understand the dominance of clade-3C.3a viruses in Israel, we examined the genetic group dominance of circulating A(H3N2) viruses in European countries reporting A(H3N2) dominance and in countries neighboring Israel (e.g., Algeria, Egypt and Tunisia). In all countries, dominance of subclade-3C.2a1 viruses was reported (data not shown). Our hypothesis is that A(H3N2) clade-3C.3a viruses have been circulating at low levels since they were identified in 2014. In support of this, at the end of the 2017–18 season (March 2018), two viruses (A/Israel/SK-1451/2018 and A/Israel/SK-1453/2018) belonging to clade-3C.3a were detected. Such clade-3C.3a viruses may have circulated at a low level in Israel during the 2018 summer, allowing their resurgence and increased circulation in the 2018–19 winter. Factors that may have contributed to this resurgence include (i) the time that elapsed since the general population was exposed to clade-3C.3a viruses, resulting in an increase in susceptible host numbers and (ii) genetic drift of clade-3C.3a viruses in the course of their low-level circulation in recent years. In the context of the first factor, prevalence of clade-3C.3a viruses varied greatly since 2014, and the patterns have been significantly different between Europe, Asia, and North America but with low-level circulation occurring between ‘outbreaks’ of clade-3C.3a viruses (Table 1).

The supposed clade-3C.3a summer circulation in Israel is supported by its observed circulation in the northern hemisphere during March–September 2018. Analysis of GISAID Epiflu^TM^ data for northern hemisphere influenza circulation during this period revealed that about 8% of the positive influenza A(H3N2) viruses belonged to clade-3C.3a. The long “post-military service” journeys, mainly to South America, Australia, and New Zealand, taken by many Israeli veterans, can reasonably explain a bulk importation of clade-3C.3a viruses into Israel during the summer season of 2018.

Regarding genetic drift, from 2014 until the winter of 2018–19, a single clade-3C.3a A/Switzerland/9715293/2013-like vaccine virus had been used in 2015–16 (Table 1). Over subsequent years, there was significant HA genetic drift, with associated HA amino acid substitutions (L3I, S91N, N144K resulting in the loss of a glycosylation sequon, F193S, R261Q, I478M and D489N) compared to the vaccine virus (Appendix A). Substitutions N144K (antigenic site A) and F193S (antigenic site B) were likely associated with the observed antigenic drift which allowed the resurgence of clade-3C.3a viruses in Israel, Europe, and North America. Host immune selection, on a global scale, is considered to be a major cause of influenza virus antigenic drift [34].

In the 2018–2019 season in Europe, Asia, and USA, A(H1N1) pdm09 viruses dominated early in the season but were displaced by A(H3N2) viruses later. Around week 8/2019 there was a peak in A(H3N2) clade-3C.3a infections in Europe, Asia, and USA, suggesting that A(H3N2) clade-3C.3a viruses effectively out-competed A(H1N1) pdm09 viruses. Specific clades/subclades of the four seasonal influenza viruses, i.e., two type A subtypes and two type B lineages, commonly co-circulate in winter outbreaks/epidemics but with one (or more) subtype/lineage predominating in different hemispheres or sometimes in both hemispheres [36]. The relative predominance of clade/subclade in each subtype/lineage is determined by the WHO and is a major factor considered when making influenza vaccine recommendations [13]. In the 2018–19 winter, ECDC and CDC reported on significant shifts in the relative proportions of three A(H3N2) clades/subclades during the course of the influenza seasons, in Europe, Asia, and USA [22,35]. Consequently, for the first time in many years, WHO postponed the announcement of the recommendation for the A(H3N2) component of influenza vaccines for the 2019–20 northern hemisphere influenza season by four weeks. The increasing prevalence of clade-3C.3a viruses in several parts of the world, including Israel, where such viruses dominated throughout the season, was taken into consideration, and A/Kansas/14/2017-like clade-3C.3a viruses were recommended for use in influenza vaccines [37].

It is well known that vaccination against influenza is currently the most effective way to prevent influenza infection and spread. Despite differences in influenza virus circulation patterns between Israel, Europe, Asia, and USA, there remain common global public health challenges crossing boundaries and affecting each continent individually and together. The unusual 2018–19 winter season with co-circulation of viruses in different A(H3N2) genetic clades/subclades, one of which (clade-3C.3a) contained viruses that were antigenically different from the vaccine virus [36], reinforces the need for universal vaccines that are effective against seasonal influenza variants within a particular subtype/lineage, regardless of genetic differences and associated antigenic differences over time.

## 5. Conclusions

In conclusion we show in this manuscript that in Israel during the 2018–19 influenza season the circulating influenza A(H3N2) viruses belonged to clade-3C.3a, while in Europe, Asia and USA they belonged to subclade-3C.2a1, but were later replaced by clade-3C.3a viruses in USA. In contrast, the vaccine A(H3N2) components components for that season, A/Singapore/INFIMH-16-0019/2016-(H3N2)-like-viruses, belonged to subclade-3C.2a1. The circulation of different influenza subtypes and clades of A(H3N2) viruses in a single season highlights the need for universal influenza vaccines.

## Figures and Tables

**Figure 1 vaccines-09-00375-f001:**
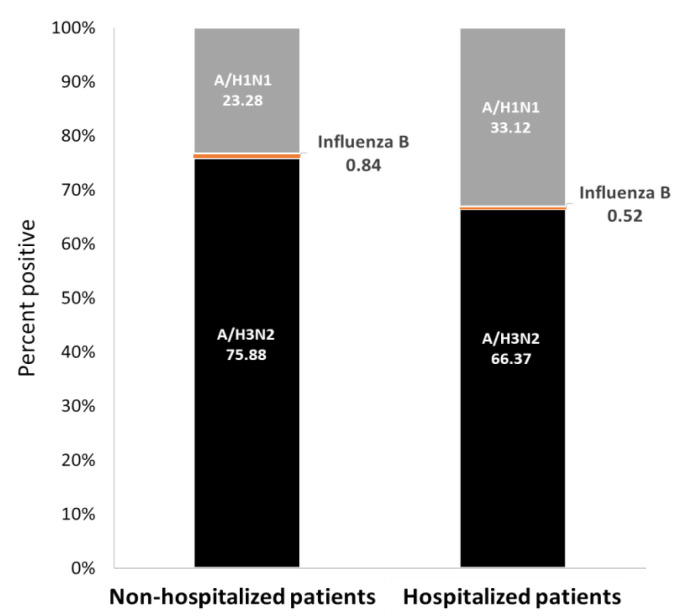
Human influenza infection cases in Israel, winter 2018–19. Data are presented as percentages of reported influenza-positive cases for non-hospitalized and hospitalized patients.

**Figure 2 vaccines-09-00375-f002:**
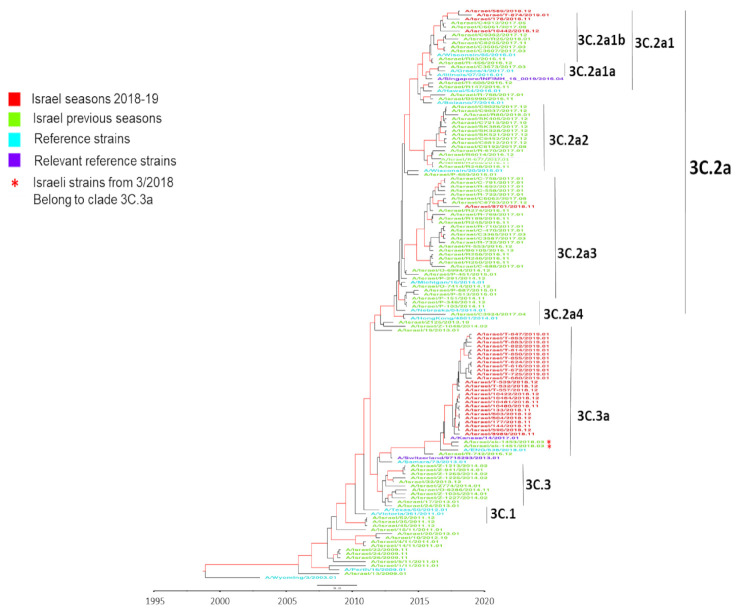
Influenza A(H3N2) hemagglutinin (HA) phylogenetic analysis using BEAST. A Bayesian maximum-clade-credibility time-scaled phylogenetic tree (BEAST) of seasonal influenza A(H3N2), generated by comparing 1701 nucleotides encoding the HA protein on a timeline of influenza samples from Israel, Europe, Asia, and USA over a 10 year period (2009 to 2019). Red indicates Israeli virus sequences from the 2018–19 season, light green indicates Israeli virus sequences from previous seasons (2009–2018), dark green indicates reference strains, light blue indicates relevant reference strains, dark blue indicates other country viruses from previous seasons (2009–2018), and purple indicates other country virus sequences from the 2018–19 season. Genetic distance is indicated by the scale bar above the year time bar. The numerals at the end of virus names (01–12) indicate the month of sample (virus) collection

**Figure 3 vaccines-09-00375-f003:**
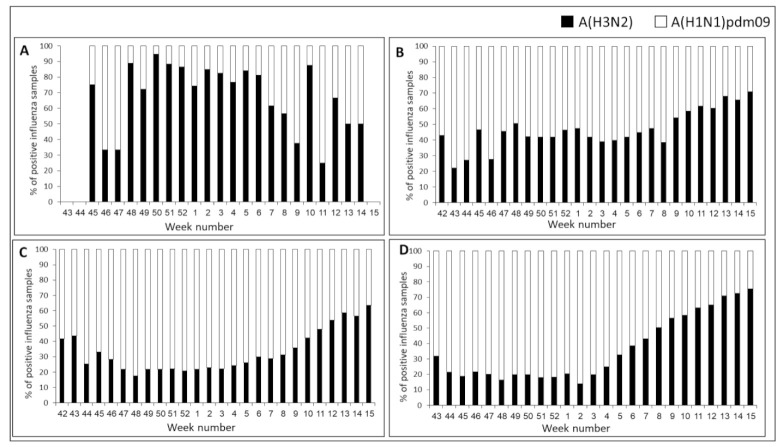
Influenza A subtype distribution in Israel, Europe, Asia, and USA, winter 2018–19. Weekly distributions (%) of influenza A(H3N2) (black bars) and A(H1N1) pdm09 (white bars) in Israel (**A**), Europe (**B**), Asia (**C**), and USA (**D**) for the 2018–2019 winter.

**Figure 4 vaccines-09-00375-f004:**
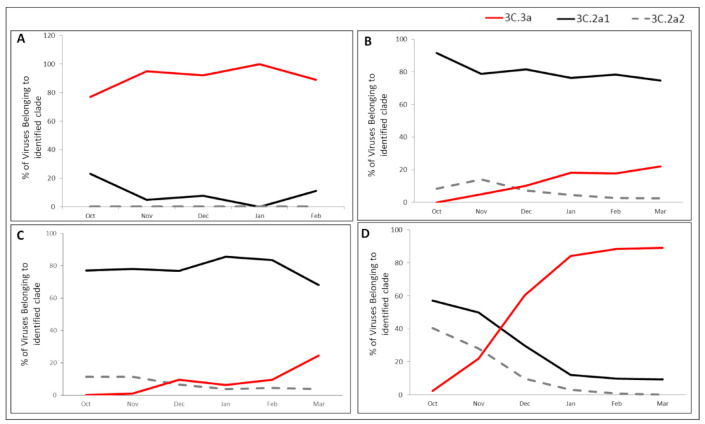
A(H3N2) clade/subclade distribution throughout the 2018–19 season in Israel, Europe, Asia, and USA. Clade distribution of the A(H3N2) viruses in Israel (**A**), Europe (**B**), Asia (**C**), and USA (**D**), in winter 2018–2019. All sequences were downloaded from the EpiFlu^TM^ database of the Global Initiative on Sharing All Influenza Data (GISAID) and classified by clade/subclade according to HA gene mutations encoding specific amino acid substitutions, based on the HA sequences of WHO vaccine viruses and reference viruses.

**Table 1 vaccines-09-00375-t001:** Circulation of A(H3N2) clade-3C.3a viruses over periods considered at the times of each WHO Vaccine Consultation Meeting (VCM) ^a^.

Vaccines Used Leading up to WHO VCM ^b^		WHO VCM ^c^	Europe ^d^			North America^d^			Asia ^d^		
NH Vaccine	SH Vaccine		No H3 HA Seq	No 3C.3a	% 3C.3a	No H3 HA Seq	No 3C.3a	% 3C.3a	No H3 HA Seq	No 3C.3a	% 3C.3a
A/Victoria/361/2011 (3C.1)	A/Texas/50/2012 (3C.1)	Sep-2014	430	12	2.8	403	62	**15.4**	553	269	**48**
A/Texas/50/2012 (3C.1)	A/Texas/50/2012 (3C.1)	Feb-2015	383	37	9.7	738	34	4.6	874	333	**38.1**
A/Texas/50/2012 (3C.1)	A/Switzerland/9715293/2013 (3C.3a)	Sep-2015	525	48	9.1	568	1	0.2	635	75	**11.8**
A/Switzerland/9715293/2013 (3C.3a)	A/Switzerland/9715293/2013 (3C.3a)	Feb-2016	123	41	**33.3**	475	12	2.5	752	60	8
A/Switzerland/9715293/2013 (3C.3a)	A/Hong Kong/4801/2014 (3C.2a)	Sep-2016	294	19	6.5	585	332	**57.0**	564	37	6.6
A/Hong Kong/4801/2014 (3C.2a)	A/Hong Kong/4801/2014 (3C.2a)	Feb-2017	818	4	0.5	690	55	8.0	1330	10	0.8
A/Hong Kong/4801/2014 (3C.2a)	A/Hong Kong/4801/2014 (3C.2a)	Sep-2017	760	6	0.8	1521	137	9.0	1180	6	0.5
A/Hong Kong/4801/2014 (3C.2a)	A/Hong Kong/4801/2014 (3C.2a)	Feb-2018	379	1	0.3	1047	9	0.9	1054	8	0.8
A/Hong Kong/4801/2014 (3C.2a)	A/Singapore/INFIMH-16-0019/2016 (3C.2a1)	Sep-2018	821	22	2.7	1032	198	**19.2**	630	5	0.8
A/Singapore/INFIMH-16-0019/2016 (3C.2a1)	A/Singapore/INFIMH-16-0019/2016 (3C.2a1)	Feb-2019	673	122	**18.1**	388	125	**32.2**	1285	56	4.4
A/Singapore/INFIMH-16-0019/2016 (3C.2a1)	A/Switzerland/8060/2017 (3C.2a2)	Sep-2019	2206	540	**24.5**	1838	1342	**73.0**	1188	608	**51.2**

^a^ For northern hemisphere (NH) recommendation meetings, viruses with collection dates from 1 September to 31 January of a given winter season are considered, while those collected between 1 February and 31 August in a single year are considered for southern hemisphere (SH) recommendation meetings. ^b^ Vaccines used in both hemispheres leading up to each VCM are indicated, with countries in tropical regions deciding which of the recommendations to use in their influenza vaccination campaigns. ^c^ Meetings held in September give recommendations for the following southern hemisphere season (e.g., September 2014 for the 2015 season), and those in February for the following NH season (e.g., February 2015 for the 2015–16 season). ^d^ For Europe, North America, and Asia, numbers of A(H3N2) HA sequences available in sequence databases, from viruses with collection dates in the periods indicated (c), at the time of each VCM, are shown with the number and % referring to clade-3C.3a viruses. Values ≥10% are shown in bold.

## Data Availability

Data is contained within the article or Appendix A.

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
