# Peer review of "Diversity in the Circulation of Influenza A(H3N2) Viruses in the Northern Hemisphere in the 2018–19 Season"

_vaccines, 2021, doi:10.3390/vaccines9040375_

Round 1

Reviewer 1 Report

The authors describe the circulating influenza strains in Israel using nasopharyngeal samples collected from outpatient and hospitalized patients and in Europe, Asia, and the USA using data reported in ECDC, CDC, and WHO databases for the same period.

The manuscript will be of interest to the journal readers in the influenza field.

I put forward some comments that can be of use to improve the manuscript. The authors will have to provide evidence for some asseverations or amend accordingly, correct typos and inadequate references.

Introduction (Background)

To not misguide readers, the authors should amend the following asseverations following the epidemiological evidence provided.

The authors write in lines 37-38, “Each season, the dominant circulating influenza subtype/lineage is usually similar worldwide [9]” and support this assertion with a unique reference to a CDC web page.

There is substantial annual geographical variability in the circulating types and subtypes and their dominance globally, see, for instance:

[1] World Health Organization (WHO). Recommended composition of influenza virus vaccines for use in the 2018–2019 northern hemisphere influenza season. Wkly Epidemiol Rec 2018;93:133–41. https://doi.org/10.1186/1750-9378-2-15.Voir.

[2] World Health Organization (WHO), Who. Recommended composition of influenza virus vaccines for use in the 2017–2018 northern hemisphere influenza season. Wkly Epidemiol Rec 2017;92:117–28. https://doi.org/10.1016/j.actatropica.2012.04.013.

[3] World Health Organization (WHO). Recommended composition of influenza virus vaccines for use in the 2016–2017 northern hemisphere influenza season. Wkly Epidemiol Rec 2016;91:121–32. https://doi.org/10.1186/1750-9378-2-15.Voir.

In line 52, they provide an inadequate reference. If they refer to the past ten years, a 2003 (Gerdil) publication makes no sense. Possibly, they intended to support their assertion with reference [16] that makes a pretty similar “ten years” statement. See last paragraph, 1st column, page 1842 in Allen JD, Ross TM. H3N2 influenza viruses in humans: Viral mechanisms, evolution, and evaluation. Hum Vaccin Immunother 2018;14:1840–7. https://doi.org/10.1080/21645515.2018.1462639.

Again, in lines 58-60, the assertion about the predominance of clade 3.C3a virus is not so clear cut as the authors assert. They should ditch the last sentence or provide evidence that supports the “consequence” that the second 2018-19 A(H3N2) was dominated by clade 3C.3a, as this was not a universal situation, see for instance [1] Mira-Iglesias A, López-labrador FX, García-Rubio J, Mengual-chuliá B. Influenza Vaccine Effectiveness and Waning Effect in Hospitalized Older Adults. Valencia Region , Spain , 2018 / 2019 Season 2021:1–17; or Figure 4 in this same manuscript.

Methods

Section 2.3 (lines 70-75) and 2.5 (lines 102-108) are duplicates; please choose one.

Actual point 2.4 fits better before point 2.7. The authors can describe everything related to the methods used to characterize Israel’s clades and follow the methods used to obtain the samples from the other regions.

Line 136, explain the groups you will compare and the rationale for comparison.

Results

Is there a typo in Figure 1?, note that the percentage of A(H1N1)pdm09 is 33% in hospitalized patients (line 144), but it is 23.2 in the Figure. This result is noted more than once in the text. Please amend it.

See lines 231-235.

It is awkward to compare the distribution observed in 600 samples with the distribution observed in tens of thousands. This sentence is an overstatement and should be tempered from comparison and contrast to “distribution of”

Figure 2. Y-axis, H1/H3 ratio %, is misleading and incorrect. It should say % of positive samples, A(H1N1)pdm09 and A(H3N2), change the numbers in the y-axis and report a prevalence ratio. Note that A, B, C, D are capital y the Figure and lower-case in the legend. This should be the same in both instances.

Comments on discussion

Line 231-235, an important point here is that the severity of A(H1N1)pdm09 has an age eschewed distribution to the young. Please introduce the point in your discussion.

Line 253 to 256. Please, support this statement with adequate evidence. Note here that there are alternative explanations, such as the egg adaptations of the egg-derived vaccine strains.

Line 327-328, please support evidence to your assertion “The unusual 2018-19 winter season with co-circulation of viruses in different 326 A(H3N2) genetic clades/subclades, one of which (clade-3C.3a) contained viruses that were 327 antigenically (reference here)”

Note that you write (lines 328-329): “universal vaccines that are effective against both seasonal influenza types or all viruses within a particular subtype/lineage, regardless of genetic differences and shifts over time.”

Be careful here not to mislead the potential reader.

The resolution of the S1 Figure is low. Provide a high-resolution s1 fig.

Author Response

Reviewer 1:

Comment: The authors describe the circulating influenza strains in Israel using nasopharyngeal samples collected from outpatient and hospitalized patients and in Europe, Asia, and the USA using data reported in ECDC, CDC, and WHO databases for the same period.

The manuscript will be of interest to the journal readers in the influenza field.

I put forward some comments that can be of use to improve the manuscript. The authors will have to provide evidence for some asseverations or amend accordingly, correct typos and inadequate references.

Response: We thank you for the evaluation of our manuscript. We made the appropriate corrections and corrected the typos.

Introduction (Background)

To not misguide readers, the authors should amend the following asseverations following the epidemiological evidence provided.

Comment: The authors write in lines 37-38, “Each season, the dominant circulating influenza subtype/lineage is usually similar worldwide [9]” and support this assertion with a unique reference to a CDC web page.

There is substantial annual geographical variability in the circulating types and subtypes and their dominance globally, see, for instance:

 [1] World Health Organization (WHO). Recommended composition of influenza virus vaccines for use in the 2018–2019 northern hemisphere influenza season. Wkly Epidemiol Rec 2018;93:133–41. https://doi.org/10.1186/1750-9378-2-15.Voir.

[2] World Health Organization (WHO), Who. Recommended composition of influenza virus vaccines for use in the 2017–2018 northern hemisphere influenza season. Wkly Epidemiol Rec 2017;92:117–28. https://doi.org/10.1016/j.actatropica.2012.04.013.

[3] World Health Organization (WHO). Recommended composition of influenza virus vaccines for use in the 2016–2017 northern hemisphere influenza season. Wkly Epidemiol Rec 2016;91:121–32. https://doi.org/10.1186/1750-9378-2-15.Voir.

Response: Thank you for this comment, we corrected our statement and added the recommended reference (lines 38-39).

Comment: In line 52, they provide an inadequate reference. If they refer to the past ten years, a 2003 (Gerdil) publication makes no sense. Possibly, they intended to support their assertion with reference [16] that makes a pretty similar “ten years” statement. See last paragraph, 1st column, page 1842 in Allen JD, Ross TM. H3N2 influenza viruses in humans: Viral mechanisms, evolution, and evaluation. Hum Vaccin Immunother 2018;14:1840–7. https://doi.org/10.1080/21645515.2018.1462639.

Response: Thank you for this comment. Indeed the reference [14] – Gerdil, was mistakenly placed in this paragraph. We removed this reference following your comment and added the suggested one (lines 56).

 Comment: Again, in lines 58-60, the assertion about the predominance of clade 3.C3a virus is not so clear-cut as the authors assert. They should ditch the last sentence or provide evidence that supports the “consequence” that the second 2018-19 A(H3N2) was dominated by clade 3C.3a, as this was not a universal situation, see for instance [1] Mira-Iglesias A, López-labrador FX, García-Rubio J, Mengual-chuliá B. Influenza Vaccine Effectiveness and Waning Effect in Hospitalized Older Adults. Valencia Region, Spain, 2018 / 2019 Season 2021:1–17; or Figure 4 in this same manuscript.

Response: Thank you for this comment; we added a reservation regarding the dominance of clade 3C.3a in the text and the suggested reference (lines 64-67).

Methods

Comment: Section 2.3 (lines 70-75) and 2.5 (lines 102-108) are duplicates; please choose one.

Response: Thank you very much. We removed the duplicate.

Comment: Actual point 2.4 fits better before point 2.7. The authors can describe everything related to the methods used to characterize Israel’s clades and follow the methods used to obtain the samples from the other regions.

Response: We thank the reviewer for this comment. we made the appropriate change.

Comment: Line 136, explain the groups you will compare and the rationale for comparison.

Response:

Thank you for this comment. Indeed, as a part of the influenza circulation description in the first part of the manuscript, we compared the influenza type distribution between hospitalized and non-hospitalized patient. This comparison revealed that the A/H3N2 was dominant in both hospitalized and non-hospitalized and that the positive A/H1N1 percentage was higher in the hospitalized patients group. (lines 161-162)

Results

Comment: Is there a typo in Figure 1?, note that the percentage of A(H1N1)pdm09 is 33% in hospitalized patients (line 144), but it is 23.2 in the Figure. This result is noted more than once in the text. Please amend it.

Response: Thank you for this comment. Indeed, there was a typo in figure 1, we apologize for this typo and we amended figure 1.

Comment: It is awkward to compare the distribution observed in 600 samples with the distribution observed in tens of thousands. This sentence is an overstatement and should be tempered from comparison and contrast to “distribution of”

Response: We changed the word "comparison" to the word "distribution" as the reviewer suggested (line 217).

Comment: Figure 2. Y-axis, H1/H3 ratio %, is misleading and incorrect. It should say % of positive samples, A(H1N1)pdm09 and A(H3N2), change the numbers in the y-axis and report a prevalence ratio. Note that A, B, C, D are capital y the Figure and lower-case in the legend. This should be the same in both instances.

Response: Thank you for this comment; we corrected the axis title according to the reviewer suggestion. We also corrected the figure legend.

Comments on discussion

Comment: Line 231-235, an important point here is that the severity of A(H1N1)pdm09 has an age eschewed distribution to the young. Please introduce the point in your discussion.

Response: Thank you for this comment; we added a short section addressing this issue (line 281-283).

Comment: Line 253 to 256. Please, support this statement with adequate evidence. Note here that there are alternative explanations, such as the egg adaptations of the egg-derived vaccine strains.

Response: Thank you for this comment, an appropriate reference was added along with an alternative explanation (line 303-311).

Comment: Line 327-328, please support evidence to your assertion “The unusual 2018-19 winter season with co-circulation of viruses in different 326 A(H3N2) genetic clades/subclades, one of which (clade-3C.3a) contained viruses that were 327 antigenically (reference here)”.

Response: A reference supporting this statement was added (line 383).

Comment: Note that you write (lines 328-329): “universal vaccines that are effective against both seasonal influenza types or all viruses within a particular subtype/lineage, regardless of genetic differences and shifts over time.”

Be careful here not to mislead the potential reader.

Response: We corrected the sentence according to the reviewer's suggestion (line 384)

Comment: The resolution of the S1 Figure is low. Provide a high-resolution s1 fig.

Response: The resolution of figure S1 was improved as the reviewer suggested.

Reviewer 2 Report

The work presented for publication is of great interest to specialists (virologists, epidemiologists and infectious disease specialists) studying the influenza virus. This article presents interesting data on the circulation of various subtypes of the human influenza virus during the epidemic in 2018-2019 in Israel. I believe that this article can be published without changes and additions.

Author Response

Reviewer 2

Comments and Suggestions for Authors

Comment: The work presented for publication is of great interest to specialists (virologists, epidemiologists and infectious disease specialists) studying the influenza virus. This article presents interesting data on the circulation of various subtypes of the human influenza virus during the epidemic in 2018-2019 in Israel. I believe that this article can be published without changes and additions.

Response: We thank the reviewer for the critical read of our manuscript.

Reviewer 3 Report

General Comment

The manuscript describes epidemiological findings for influenza infections in Israel between October 2018 and March 2019. An in-depth analysis of HA sequences from samples collected from both hospitalized and non-hospitalized patients was performed to compare data with Europe, Asia, and the USA.

  1. The "Material and Methods" and the "Results" sections should be reorganized by separating the analysis of samples and data carried out in Israel from the comparison with data from other countries.
  2. The "Discussion" section should be improved. More details about the comparison of data among the considered countries should be reported. Moreover, the public health impact of the mismatch with the influenza vaccine should be better described. For instance, it could be taken into account the reported vaccine effectiveness of the influenza vaccine for the different countries.
  3. Line 49. It should be specified the whole composition for the influenza vaccine for the Northern hemisphere in the considered season
  4. Paragraphs 2.3 and 2.6 seem to be the same
  5. In the title of paragraph 2.4, the word "sample" should be changed to "data", since no samples from Europe, Asia, or the USA were analyzed by the authors
  6. Line 146. The sentence should be modified in "was significantly higher in hospitalized patients"

Author Response

Reviewer 3

Comments and Suggestions for Authors

General Comment

The manuscript describes epidemiological findings for influenza infections in Israel between October 2018 and March 2019. An in-depth analysis of HA sequences from samples collected from both hospitalized and non-hospitalized patients was performed to compare data with Europe, Asia, and the USA.

Comment: The "Material and Methods" and the "Results" sections should be reorganized by separating the analysis of samples and data carried out in Israel from the comparison with data from other countries.

Response: Thank you for this comment, we changed the order of the materials and methods and the results section according to the reviewer's suggestion.

Comment:  The "Discussion" section should be improved. More details about the comparison of data among the considered countries should be reported. Moreover, the public health impact of the mismatch with the influenza vaccine should be better described. For instance, it could be taken into account the reported vaccine effectiveness of the influenza vaccine for the different countries.

Response: Thank you for this comment, we expended the discussion and added more relevant information regarding the vaccine effectiveness in the 2018-19 winter in different countries (lines 303-311).

Comment:  Line 49. It should be specified the whole composition for the influenza vaccine for the Northern hemisphere in the considered season

Response: We added the 2018-19 whole vaccine composition for the northern hemisphere as the reviewer suggested (line 50-53).

Comment:  Paragraphs 2.3 and 2.6 seem to be the same

Response: Thank you very much. We removed the duplicate section.

Comment:  In the title of paragraph 2.4, the word "sample" should be changed to "data", since no samples from Europe, Asia, or the USA were analyzed by the authors

Response: Thank you for this comment. we changed the title according to the reviewer's suggestion (line 108).

Comment:  Line 146. The sentence should be modified in "was significantly higher in hospitalized patients"

Response: We changed the sentence according to the reviewer's suggestion (line 170).

Round 2

Reviewer 1 Report

The authors have amended various errors in their previous manuscript, such as Figure 1 misplaced information that reversed the data on critical results, misplaced references and duplicated paragraphs and sections. 

The manuscript, in its revised form, has to be substantially improved.

The abstract is difficult to read, with unconnected sentences. The authors should make an effort to make the reading of this important piece of their manuscript easy an fluid for the benefit of potential readers.

Introduction

Line 37-39. The authors maintain the "usual similar" homogeneity of circulating subtype/lineage influenza viruses globally and insist on supporting this point with reference (9), adding now reference (15), omitting to direct the reader to any particular season, here I again recommend to cite the WHO publications with the 2017-2018 season and 2018-2019 season recommended vaccine composition. They can use reference 15 as used in line 50, but even here, this all-encompassing reference strategy is not adequate. 

Note that they make the point for a twice a year recommendation for vaccine composition in lines 44-45 (reference 14 here) and repeat it in lines 48 to 50 (reference 15, here). Given the focus is mainly on the northern hemisphere, needs this point to be presented twice? 

I would recommend that they ditch entirely the assertion on lines 37-39. If they consider this critical point of their manuscript, use a full-fledged epidemiological evidence-loaded reference: World Health Organization (WHO). Recommended composition of influenza virus vaccines for use in the 2018–2019 northern hemisphere influenza season. Wkly Epidemiol Rec 2018;93:133–52.

Line 53, use this same reference (World Health Organization (WHO). Recommended composition of influenza virus vaccines for use in the 2018–2019 northern hemisphere influenza season. Wkly Epidemiol Rec 2018;93:133–52) to support lines 50-53.

Lines 67-68. This is an unsupported assertion, as the picture was less glossy, in fact: 

"The majority of A(H3N2) viruses collected from Septem- ber 2018 to January 2019 belonged to the phylogenetic subclade 3C.2a1b; however, the number of clade 3C.3a viruses has increased substantially since November 2018 in several geographic regions. ( World Health Organization (WHO). Recommended composition" "of influenza virus vaccines for use in the 2019–2020 northern hemisphere influenza season. Wkly Epidemiol Rec 2019;94:141–60.)"

So you should modify this sentence or, if it was intended for Israel, say it so:

"Consequently, In Israel, most of the A(H3N2) viruses in circulation differed from the subclade 3C.2a1, A/Singapore/INFIMH-16-0019/2016-(H3N2)-like, vaccine viruses. "

I  suggest that you close the introduction specifying: which was the aim of the study?

Note that this important piece of information is missing, and keeps the reader asking himself, which is the purpose of this text?

Methods

Line 76-81. Was any systematic sampling approach? Please, clarify in the text.

Results

Lines 142, if the information provided in Figure 1, 75% should be 76%.

Line 150 3.3?

Line 154, Figure 3, should be in this revised manuscript Figure 2, following the order that figures are mentioned in the text.

Line 154, demonstrated, should be "showed"  or "was compatible."

Line 174, 3.2?

Line 175-185, Figure 2 should be named now Figure 3. As the authors opted to capitalize the letters identifying the graphs in the figures, we recommend following the same rule in the text. 

Line 197, Fig. 4a, should be Fig. 4A, and keep this capital nomination constant in all the text, given that the authors choose to keep the capital letters on the previous submission figure 4. 

Discussion

The first paragraph, lines 223-226, is out of place.

The discussion should begin with paragraph lines 227-231.

Line 292, antigenic drift? or genetic drift? Note than in table 1, you do not present antigenicity data, but clade prevalence information.

Line 298. Please. be accurate when mixing antigenicity data with genetic data.  Is antigenic drift the cause of genetic changes? Drop it.

Author Response

The authors have amended various errors in their previous manuscript, such as Figure 1 misplaced information that reversed the data on critical results, misplaced references and duplicated paragraphs and sections.

Comment: The manuscript, in its revised form, has to be substantially improved.

Response: We thank the reviewer for the evaluation of our manuscript. We corrected the manuscript in complete accordance with the reviewers' comments

Comment: The abstract is difficult to read, with unconnected sentences. The authors should make an effort to make the reading of this important piece of their manuscript easy an fluid for the benefit of potential readers.

Response Thank you for this comment. We rewrote the abstract. See lines 15-26.

Introduction

Comment: Line 37-39. The authors maintain the "usual similar" homogeneity of circulating subtype/lineage influenza viruses globally and insist on supporting this point with reference (9), adding now reference (15), omitting to direct the reader to any particular season, here I again recommend to cite the WHO publications with the 2017-2018 season and 2018-2019 season recommended vaccine composition. They can use reference 15 as used in line 50, but even here, this all-encompassing reference strategy is not adequate.

Response: We added the requested WHO publications for these two years. See lines 51-57.

Comment: Note that they make the point for a twice a year recommendation for vaccine composition in lines 44-45 (reference 14 here) and repeat it in lines 48 to 50 (reference 15, here). Given the focus is mainly on the northern hemisphere, needs this point to be presented twice?

Response: We deleted the point mentioned in lines 44-45.

Comment: I would recommend that they ditch entirely the assertion on lines 37-39. If they consider this critical point of their manuscript, use a full-fledged epidemiological evidence-loaded reference: World Health Organization (WHO). Recommended composition of influenza virus vaccines for use in the 2018–2019 northern hemisphere influenza season. Wkly Epidemiol Rec 2018;93:133–52.

Response: We deleted this sentence.

Comment: Line 53, use this same reference (World Health Organization (WHO). Recommended composition of influenza virus vaccines for use in the 2018–2019 northern hemisphere influenza season. Wkly Epidemiol Rec 2018;93:133–52) to support lines 50-53.

Response: We cite the suggested reference (Ref 18), See line 60

Comment: Lines 67-68. This is an unsupported assertion, as the picture was less glossy, in fact:

"The majority of A(H3N2) viruses collected from Septem- ber 2018 to January 2019 belonged to the phylogenetic subclade 3C.2a1b; however, the number of clade 3C.3a viruses has increased substantially since November 2018 in several geographic regions. ( World Health Organization (WHO). Recommended composition" "of influenza virus vaccines for use in the 2019–2020 northern hemisphere influenza season. Wkly Epidemiol Rec 2019;94:141–60.)"

Response: We modified the sentence as requested.  Please see lines (Ref 19) 61-65

Comment: So you should modify this sentence or, if it was intended for Israel, say it so:

"Consequently, In Israel, most of the A(H3N2) viruses in circulation differed from the subclade 3C.2a1, A/Singapore/INFIMH-16-0019/2016-(H3N2)-like, vaccine viruses. "

Response: We modified the sentence as above. Please see lines 75-77

Comment: I  suggest that you close the introduction specifying: which was the aim of the study?

Note that this important piece of information is missing, and keeps the reader asking himself, which is the purpose of this text?

Response:  We close the introduction with the aim of the study as suggested. Please see line 66-67

Methods

Comment: Line 76-81. Was any systematic sampling approach? Please, clarify in the text.

Response: We added the reasons for sampling approach as suggested. Please see lines 83-89-92

Results

Comment: Lines 142, if the information provided in Figure 1, 75% should be 76%.

Response: We corrected the information. Please see line 153

Comment: Line 150 3.3?

Response: we corrected the numbering. Please see line 161

Comment: Line 154, Figure 3, should be in this revised manuscript Figure 2, following the order that figures are mentioned in the text.

Response: We corrected the figures numbering.

Comment: Line 154, demonstrated, should be "showed"  or "was compatible."

Response: We corrected the phrase according to the reviewer suggestion. Please see line 165

Comment: Line 174, 3.2?

Response: we corrected the numbering. Please see line 161

Comment: Line 175-185, Figure 2 should be named now Figure 3. As the authors opted to capitalize the letters identifying the graphs in the figures, we recommend following the same rule in the text.

Response: We corrected the figures numbering and changed the text to capital letters. Please see line 199-201

Comment: Line 197, Fig. 4a, should be Fig. 4A, and keep this capital nomination constant in all the text, given that the authors choose to keep the capital letters on the previous submission figure 4.

Response: We changed the text to capital letters. Please see line 208-226..

Discussion

Comment: The first paragraph, lines 223-226, is out of place.

The discussion should begin with paragraph lines 227-231.

Response: We changed the paragraph according to the reviewer suggestion.

Comment: Line 292, antigenic drift? or genetic drift? Note than in table 1, you do not present antigenicity data, but clade prevalence information.

Response: We change the word to genetic. Please see line 296.

Comment: Line 298. Please. be accurate when mixing antigenicity data with genetic data.  Is antigenic drift the cause of genetic changes? Drop it

Response: Changed to genetic. Please see line 309.

Reviewer 3 Report

The manuscript does not require further modification

Author Response

There were no comments.

Round 3

Reviewer 1 Report

The authors have answered all questions raised in previous reviews and revised the manuscript. I have no additional comments.